# Primal-Dual Mesh Convolutional Neural Networks

**Francesco Milano**[*]
ETH Zurich, Switzerland
fmilano@student.ethz.ch

**Antonio Loquercio**
Robotics and Perception Group
University of Zurich, Switzerland
loquercio@ifi.uzh.ch

**Antoni Rosinol**
SPARK Lab
MIT, USA
arosinol@mit.edu

**Davide Scaramuzza**
Robotics and Perception Group
University of Zurich, Switzerland
sdavide@ifi.uzh.ch

**Luca Carlone**
SPARK Lab
MIT, USA
lcarlone@mit.edu

## Abstract

Recent works in geometric deep learning have introduced neural networks that allow performing inference tasks on three-dimensional geometric data by defining convolution –and sometimes pooling– operations on triangle meshes. These methods, however, either consider the input mesh as a graph, and do not exploit specific geometric properties of meshes for feature aggregation and downsampling, or are specialized for meshes, but rely on a rigid definition of convolution that does not properly capture the local topology of the mesh. We propose a method that combines the advantages of both types of approaches, while addressing their limitations: we extend a primal-dual framework drawn from the graph-neural-network literature to triangle meshes, and define convolutions on two types of graphs constructed from an input mesh. Our method takes features for both edges and faces of a 3D mesh as input, and dynamically aggregates them using an attention mechanism. At the same time, we introduce a pooling operation with a precise geometric interpretation, that allows handling variations in the mesh connectivity by clustering mesh faces in a task-driven fashion. We provide theoretical insights of our approach using tools from the mesh-simplification literature. In addition, we validate experimentally our method in the tasks of shape classification and shape segmentation, where we obtain comparable or superior performance to the state of the art.

## 1 Introduction

The development of deep-learning tools that operate on three-dimensional triangular meshes has recently received an increasing attention by the computer graphics, vision, and machine learning communities, due to the availability of large 3D datasets with semantic annotations [1, 2] and to the expressiveness and efficiency of meshes in representing non-uniform, irregular surfaces. Compared to alternative representations often used for 3D geometry, such as voxels and point clouds, that scale poorly to high object resolutions [3], meshes allow representing structure more adaptively and compactly [4]; at the same time, they naturally provide connectivity information, as opposed to point clouds, and lend themselves to simple, commonly used rendering and processing algorithms [5, p. 14]. However, meshes have both a *geometric* and a *topological* component [6, p. 10], represented respectively by the vertices and by the connectivity of edges and faces [5, p. 19]; while the geometrical variability of the underlying surface encodes essential semantic information, the randomness in the discretization of the surface (which can include, e.g., variations in tessellation, isotropy, regularity, *level-of-detail* [5]) is to a certain extent not informative and independent of the

---

[*]Work performed while at MIT.

shape identity [7]. This is particularly relevant in the context of shape understanding and semantic-based object representation, where mesh processing requires abstracting from the low-level mesh elements to a higher-level structure-aware representation of the shape [8–10].

Handling 3D geometric data represented as meshes and performing deep learning tasks on them relates to the field of geometric deep learning [11], that attempts to generalize tools from convolutional neural networks (CNNs) to non-Euclidean domains, including graphs and manifolds. Existing methods define specific *convolution* and (optionally) *pooling/unpooling* operations to operate on meshes [12]. We identify two main types of approaches: on the one hand, methods that process the input mesh as a graph [13–16], thus not exploiting important geometrical properties of meshes [17]; on the other hand, *ad-hoc* methods, which design convolution and pooling operations using geometric properties of triangular meshes. However, both types of approaches usually implement a form of *non-dynamic* feature aggregation. Indeed, they learn and apply shared isotropic kernels to all the vertices/edges of the mesh and use a weighting scheme that does not depend on the region of the mesh on which the operation is applied. This is a limiting factor, considering that meshes can exhibit large variations in the density and shape of their faces. Furthermore, graph-based methods often do not perform pooling, or they implement it through generic graph clustering algorithms [18], that do not exploit the mesh geometric characteristics. In contrast, the ad-hoc methods that define mesh-specific pooling operations [12] make strong assumptions on the mesh local topology, limiting the number of mesh elements which can be pooled.

**Contributions.** To address the limitations of the approaches above, we propose a method that combines a graph-neural-network framework with mesh-specific insights from ad-hoc approaches. Our method, named PD-MeshNet, is the first to perform dynamic, attention-based feature aggregation while also implementing a task-driven, mesh-specific pooling operation. Motivated by the inherent duality between topology and geometry in meshes, we build on the primal-dual graph convolutional framework of [19], and extend it to triangular meshes: starting from an input mesh, we construct two types of graphs, that allow assigning features to both edges and faces of a 3D mesh. The method enables the implementation of *dynamic* feature aggregation –where the relevance of each neighbor in the convolution operation is learned using an attention mechanism [20]– and at the same time allows to learn richer and more complex features [19]. On the other hand, we show that *ad-hoc* methods, such as [12], can be interpreted as an instantiation of our method where only a single graph is considered. Similarly to [12], we also introduce a task-driven pooling operation specific for meshes. A unique feature of our pooling operation, however, is its geometrical interpretation: by collapsing graph edges based on the associated attention coefficients, PD-MeshNet learns to form clusters of faces in the mesh, allowing both to downsample the number of features in the network and to abstract the computation from the low-level, noise-prone mesh elements, to larger areas in the shape. In addition, our pooling operation does not require assumptions on the topological type of the meshes nor has the topological limitations of [12]. We evaluate our method in the tasks of mesh classification and segmentation, where we show results comparable or superior to state-of-the-art approaches. Our code is publicly available at `https://github.com/MIT-SPARK/PD-MeshNet`.

**Notation.** In the following, we assume the reader to be familiar with basic notions from graph theory (*e.g.*, graph embedding, $k$-regularity [21, 22]) and meshes (*e.g.*, 1-ring neighborhood, boundary edge/face [5, 6]). We denote the vertices of a generic triangle mesh $\mathcal{M}$ with lowercase letters (*e.g.*, $a$), and its faces with uppercase letters (*e.g.*, $A$). We refer to the set of all the faces of the mesh as $\mathcal{F}(\mathcal{M})$, and we denote the set of the faces adjacent to a generic face $A \in \mathcal{F}(\mathcal{M})$ as $\mathcal{N}_A$. For simplicity and comparability to [12], we assume the mesh $\mathcal{M}$ to be *(edge-)manifold*[2]. However, as we show in the Supplementary Material, our framework allows to relax this assumption and process meshes of any topological type.

## 2   Related Work

**Graph-based methods.** Related to our approach are works that consider meshes as graph-structured data, which is processed by one of the existing variants of graph convolutional networks (GCNs) [23–25, 20, 19]. Some of the methods based on GCNs are specifically designed to work on manifold meshes, whose connectivity allows to define convolution on patches around mesh vertices represented

in an intrinsic coordinate system [13–15]. The above approaches, however, apply shared isotropic kernels to the mesh vertices according to a weighting scheme that is independent of the position on the surface. Therefore, they do not adapt to local variations in the regularity and isotropy of mesh elements. Partially addressing this problem, Verma et al. [16] propose a method that dynamically learns the correspondence between filter weights and neighboring nodes. However, their method implements the pooling operation through a generic graph clustering algorithm [18], which is not aware of the geometry of the mesh. Other graph-based approaches [26] define pooling using classic mesh-simplification techniques for surface approximation [27]. However, the latter algorithm, which approximates the mesh surface by minimizing a geometric error, is not necessarily optimal for the downstream task (*e.g.*, classification or segmentation). In contrast, our approach uses a pooling operation which is both mesh-specific and task-driven. Finally, the recent work of Schult et al. [28] made a step towards addressing dynamic feature aggregation in meshes by proposing to combine a geodesic-based vertex convolution with another convolution on vertices based on Euclidean distance. However, this approach implements pooling through classic mesh-simplification methods that minimize non-task-driven geometric errors [28].

**Ad-hoc methods.** A second class of methods defines convolution and/or pooling by using *ad-hoc* mesh properties [29, 12, 30–33]. Tatarchenko et al. [30] process mesh surfaces with standard 2D convolutions, which operate on feature maps defined on local tangent planes. Similarly, Huang et al. [31] use standard CNNs to extract features from high-resolution texture signals. Feng et al. [29] design a mesh-specific convolution that aggregates spatial and structural features of mesh faces. Hanocka et al. [12] define convolution over edges of the input mesh, which are assumed to be edge-manifold. This assumption allows defining a symmetric convolution operation, which aggregates features from the 1-ring neighborhood of each edge. In addition, it allows pooling according to a task-driven version of the classical *edge-collapse* operation from the mesh-simplification literature [34]. The method of Lim et al. [32], further developed by Gong et al. [33], implements a convolution operation based on *spirals* that are defined around each mesh vertex; this approach, however, requires all the meshes in the dataset to have a similar, fixed, topology. The remaining aforementioned methods, similarly to graph-based approaches, either do not perform *dynamic* aggregation of features [12, 29] or do not provide a mesh-specific downsampling operation [30, 31]. The few exceptions, which define an ad-hoc mesh-pooling operation, *e.g.* [12], make strong assumptions on the topology of the input mesh, *de facto* limiting the number of mesh elements which can be pooled. In contrast, our method performs dynamic feature aggregation through an attention mechanism, and exploits the latter to define a novel pooling operation, tailored to the mesh and the task, which is not limited by the mesh topology.

## 3  PD-MeshNet

This section introduces the building blocks of our method, and in particular the instantiation of the primal-dual graph (Section 3.1), convolution (Section 3.2), and task-driven pooling/unpooling operations (Sections 3.3-3.4).

### 3.1  Converting 3D Meshes to Graphs

Given an input mesh $\mathcal{M}$, PD-MeshNet constructs a primal and a dual graph (Fig. 1).

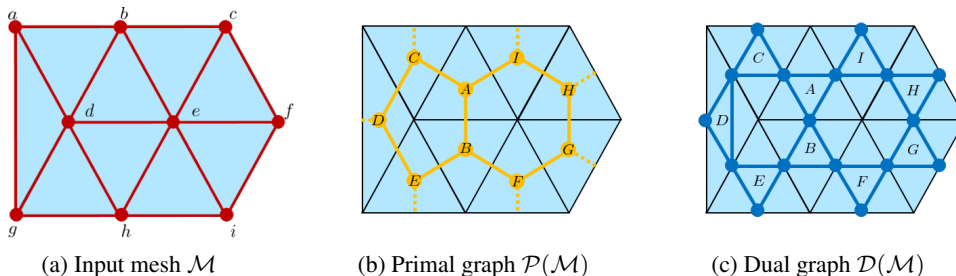

(a) Input mesh $\mathcal{M}$      (b) Primal graph $\mathcal{P}(\mathcal{M})$      (c) Dual graph $\mathcal{D}(\mathcal{M})$

Figure 1: Primal-dual graphs associated to an input mesh in PD-MeshNet. Vertices and faces of the triangle mesh are denoted respectively by lowercase and uppercase letters.

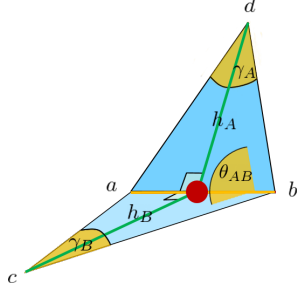

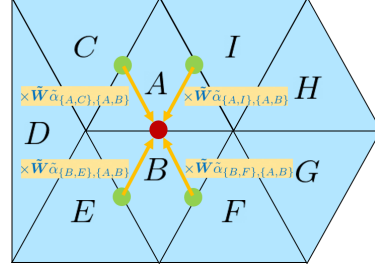

Figure 2: Features of a generic dual node $\{A, B\}$: dihedral angle $\theta_{AB}$, internal angles $\gamma_A$ and $\gamma_B$, edge-to-height ratios $\|ab\|/h_A$ and $\|ab\|/h_B$.

Figure 3: Aggregation of the neighboring features for a generic dual node $\{A, B\}$ (in red).

The **Primal Graph** of a mesh $\mathcal{M}$ is an undirected graph having a node for each face of $\mathcal{M}$ and an edge between two nodes if the corresponding faces are adjacent in $\mathcal{M}$ (Fig. 1b). We denote our primal graph as $\mathcal{P}(\mathcal{M})$. Contrary to related work [13–17], which builds a graph over the mesh vertices, our primal graph operates over the faces of the mesh. The graph built on the mesh vertices – that we denote $\mathcal{G}(\mathcal{M})$, and that is visually similar to the one in Fig. 1a – is sometimes called *mesh graph* [3, 16, 17]. On the other hand, a representation akin to the proposed primal graph has been also referred to as *simplex mesh* in related work [35]. By construction, our primal graph $\mathcal{P}(\mathcal{M})$ is topologically dual of $\mathcal{G}(\mathcal{M})$ [35, 36]. The advantage of our primal graph is that it allows defining pooling/unpooling operations that can be easily interpreted in terms of clustering of the mesh faces. For each mesh face $A \in \mathcal{F}(\mathcal{M})$, we assign to the corresponding node in the primal graph –which we also denote as $A$– the feature $f_A$, defined as the ratio between the area of face $A$ and the sum of the areas of all the faces in $\mathcal{F}(\mathcal{M})$.

The **Dual Graph** of a mesh $\mathcal{M}$ is a graph having a node for each edge $e \in \mathcal{M}$, and an edge connecting two nodes where the corresponding mesh edges are adjacent to a face in $\mathcal{M}$ (Fig. 1c). Interestingly, our dual graph captures the model used in the ad-hoc method of Hanocka et al. [12], which, however, does not explicitly use a graph-based representation. Indeed, as we show in the Supplementary Material, for an edge-manifold triangle mesh $\mathcal{M}$, our dual graph $\mathcal{D}(\mathcal{M})$ (i) is the *line graph* of $\mathcal{P}(\mathcal{M})$ and (ii) is the *medial graph* of $\mathcal{G}(\mathcal{M})$. The dual graph allows performing feature aggregation among edges in a 1-ring neighborhood, which for a manifold triangle mesh is 4-regular (a property exploited in [12]). With such geometric interpretation we take a step forward to bridging the gap between graph-based and ad-hoc methods for mesh processing. This allows our approach to benefit from their respective characteristics.

**Dual Graph Construction.** For each pair of adjacent mesh faces $A, B \in \mathcal{F}(\mathcal{M})$, a *single* node $\{A, B\}$ is created in the dual graph, with undirected edges connecting it to the nodes of the form $\{A, M\}, M \in \mathcal{N}_A \backslash \{B\}$ and $\{B, N\}, N \in \mathcal{N}_B \backslash \{A\}$ (cf. Fig. 1c). However, this is not the only viable option to generate the dual graph of the mesh: there are three admissible configurations of the dual graph, which differ in the number of nodes corresponding to each mesh edge and in the directedness of the graph edges. We experimentally noticed minor performance differences by changing the dual graph configuration. In the Supplementary Material we provide more details about these configurations as well as an ablation study of their impact on performance.

**Dual Graph Features.** We assign to each node $\{A, B\}$ of the dual graph the same type of geometric features as in [12]. Specifically, we use the following features: (i) the dihedral angle $\theta_{AB}$ between faces $A$ and $B$, (ii) the ratios between the edge shared by $A$ and $B$ and the heights of the two faces with respect to the shared edge (edge-to-height ratios), (iii) the internal angles of the two faces. A geometric illustration of these features is presented in Fig. 2.

## 3.2 Convolution

After converting an input 3D mesh to a pair of primal and dual graphs as defined above, we perform the convolution operation using the method of Monti et al. [19], which was previously applied only to graphs from standard graph benchmark datasets [37, 38].

A *primal-dual convolutional layer* consists in the application of two alternating convolution operations on the dual and on the primal graph. In particular, the dual convolutional layer is a graph attention network (GAT) [20]: for a generic dual node $\{A, B\}$ the layer outputs a feature $\tilde{f}'_{\{A, B\}}$ obtained

by aggregating the features of its neighboring nodes $\{A, M\}, M \in \mathcal{N}_A \backslash \{B\}$ and $\{B, N\}, N \in \mathcal{N}_B \backslash \{A\}$, transformed through a shared learnable kernel $\tilde{W}$. The key property of the approach is that the aggregation is further weighted through *attention coefficients* defined on the edges, $\{A, M\} \rightarrow \{A, B\}$ and $\{B, N\} \rightarrow \{A, B\}$ (Fig. 3); for a generic neighboring node $\{A, M\}, M \in \mathcal{N}_A \backslash \{B\}$, the attention coefficient $\tilde{\alpha}_{\{A,M\},\{A,B\}}$ associated to the edge $\{A, M\} \rightarrow \{A, B\}$ is computed as a variation of the softmax function of the features of $\{A, B\}$, of $\{A, M\}$, as well as the other neighboring nodes of $\{A, B\}$, parameterized by a learnable attention parameter $\tilde{a}$. Similarly, the primal convolution consists of a GAT, with a shared learnable kernel $W$ and *primal attention coefficients* $\alpha_{M,A}, M \in \mathcal{N}_A$ defined, for each primal node $A$, on the incoming edges of the primal graph. However, since every primal edge has a corresponding node in the dual graph, primal attention coefficients are computed from the dual features; in particular, the attention coefficient $\alpha_{B,A}$, associated to the generic primal edge $B \rightarrow A$, with $B \in \mathcal{N}_A$, is obtained through a variation of the softmax function of the features of the dual node $\{A, B\}$ and of all the dual nodes of the form $\{A, M\}, M \in \mathcal{N}_A$, parameterized by a learnable attention parameter $a$. Similarly to [20], multiple versions - also called *heads* - of both the attention parameters $\tilde{a}$ and $a$ can be used.

## 3.3 Pooling

Our pooling operation consists in an *edge contraction* [21, p. 264] performed in the primal graph $\mathcal{P}(\mathcal{M})$. While edge contraction has recently been used also in the graph-neural-network literature to implement pooling on generic graphs [39–41], the unique feature of our approach is the geometric interpretation that this operation has in our framework. Indeed, it is easy to see that an edge contraction in the primal graph $\mathcal{P}(\mathcal{M})$ corresponds to *merging faces of the mesh* $\mathcal{M}$ (cf. Fig. 4). This idea was exploited in a classical work in mesh simplification for the purpose of forming *clusters of faces* in meshes [42]. However, while in [42] the edges to be contracted were selected to minimize a

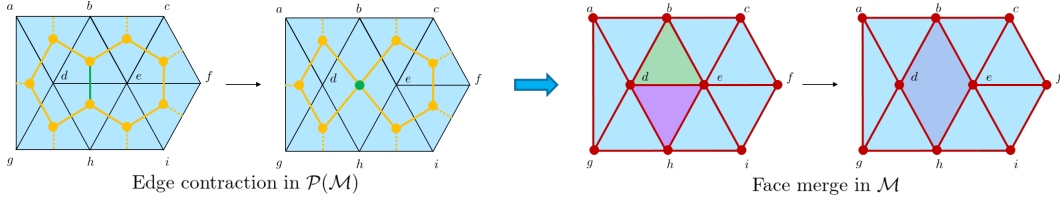

Edge contraction in $\mathcal{P}(\mathcal{M})$        Face merge in $\mathcal{M}$

Figure 4: Our pooling operation consists in an attention-driven edge contraction in the primal graph $\mathcal{P}(\mathcal{M})$, which corresponds to merging faces of the mesh $\mathcal{M}$.

geometric and task-independent error, our approach lets the network *learn* what edges of $\mathcal{P}(\mathcal{M})$ should be contracted, using the associated primal attention coefficients as a criterion. The key idea of our method is that the attention coefficients between two adjacent primal nodes should encode how *relevant* the information flow between the two corresponding faces of the mesh is, with respect to the task for which the network is trained. This way, by stacking multiple pooling layers PD-MeshNet is able to aggregate feature information over faces and form larger clusters of faces optimized for the task at hand.

More specifically, given two adjacent faces $A, B \in \mathcal{F}(\mathcal{M})$, we choose whether to contract the edge between the corresponding primal nodes based on the sum of the attention coefficients along the two directions $A \rightarrow B$ and $B \rightarrow A$, i.e.,

$$\alpha_{A,B} + \alpha_{B,A}. \tag{1}$$

In case multiple attention heads are used, the values (1) from the different heads are averaged.

Each pooling layer contracts the $K$ edges with largest cumulative coefficients (1), where $K$ is a user-specified parameter[3]; the edges are contracted in parallel. At the output of each pooling layer, the dual graph is efficiently reconstructed on the basis of the pooled primal edges, so as to represent the line graph of the new primal graph[4] (cf. Fig. 5).

We observe that the top-$K$ pooling approach described above allows any two adjacent primal edges to be pooled at the same time, if they are both among the $K$ edges with the largest quantity (1); indeed,

in general the operation merges $n \geq 2$ primal nodes $A_1, A_2, \cdots, A_n$ whose corresponding faces in the mesh form a *triangle fan* [43] into a single primal node, that we denote as $A_1 A_2 \cdots A_n$.

The feature of the new primal node $A_1 A_2 \cdots A_n$ is determined by summing the features of its constituent nodes $A_1, A_2, \cdots, A_n$, *i.e.*, $\boldsymbol{f}_{\boldsymbol{A_1} \boldsymbol{A_2} \cdots \boldsymbol{A_n}} \coloneqq \sum_{i=1}^{n} \boldsymbol{f}_{\boldsymbol{A_i}}$. We also considered using averaging as the aggregation function, but empirically found sum-based aggregation to perform slightly better (cf. Supplementary Material for a more detailed comparsion).



Figure 5: Left side: An edge (shown in green) in the primal graph (depicted in yellow) is contracted. Right side: the dual graph is updated so as to match the line graph of the new primal graph. As an example, shown in red is the new dual node $\{AB, C\}$ with its neighborhood in the updated dual graph.

In general, it is possible that face clusters corresponding to two new adjacent primal nodes shared more than one edge before pooling. Consider, as an example, the case in which the nodes $A, B$ and the nodes $C, D, E$ in Fig. 5 were merged into two primal nodes $AB$ and $CDE$, collapsing the edges between $A$ and $B$, $C$ and $D$, and $D$ and $E$: two primal edges would connect the new primal nodes, corresponding to the dual nodes $\{A, C\}$ and $\{B, E\}$ of the original graph.

As shown in Fig. 6, whenever this happens, the two (or more) dual nodes are merged into a single dual node ($\{AB, CDE\}$ in the example). Similarly to the case of primal nodes, this node is assigned as feature the sum of the features of its constituting nodes. Two other cases are possible for dual graphs when a primal edge is contracted: (i) a dual node is removed from the graph, when the corresponding primal edge is contracted, (ii) a dual node is kept in the graph with the same feature, if it is the only dual node that corresponds to an edge between two new primal nodes.

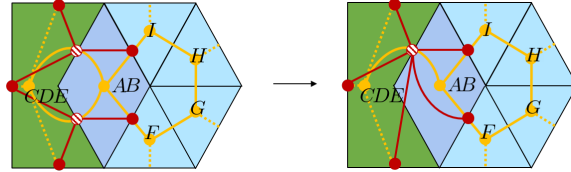

Figure 6: Collapsing primal edges $A$-$B$, $C$-$D$, and $D$-$E$ in the left side of Fig. 5 causes two existing dual nodes (striped on the left side) to correspond to primal edges between the same two new primal nodes $AB$ and $CDE$. The two dual nodes get merged into a single one (striped on the right side).

We finally note that the edge collapse operation used in [12] suffers from a topological limitation: collapsing a mesh edge which is adjacent to a valence-3 vertex is not allowed, as it would cause the formation of a non-manifold edge [44], breaking the required assumption of edge manifoldness (cf. Sec. C.1 of the Supplementary Material). In practice, this limits the number of edges that can be pooled, and consequently poses a restriction to the resolution that can be reached through the downsampling operation. On the contrary, our pooling method has no such limitations (in principle, it would be possible to obtain a single face cluster for each connected component in the mesh [42]), and has the further advantage of allowing at any time to map an element (face) in the original mesh to the corresponding face cluster in the simplified mesh.

## 3.4 Unpooling

As we show in the next section, the experiments performed on mesh segmentation tasks rely on an encoder-decoder architecture, and thus require implementing an unpooling operation. To achieve this, we simply store the connectivity of the primal and dual graphs at the input of each pooling layer, and we use look-up operations to restore it in the unpooling layers, which are in 1-to-1 correspondence with the pooling layers. Therefore, the operation maps larger face clusters to smaller ones; both the primal and the dual nodes associated to the smaller clusters are assigned the same features as the corresponding nodes in the larger clusters. Due to the fact that a pooling operation may remove some dual nodes (cf. Section 3.3), in general some nodes in the dual graph outputted by the unpooling layer will not have a corresponding node in the input dual graph; we learn a single parameter vector that we assign as feature to all such nodes.

# 4 Experiments

We evaluate PD-MeshNet on the tasks of mesh classification and mesh segmentation. On these tasks and on several datasets, we outperform start-of-the-art methods. In all the experiments we use Adam algorithm [45] for optimization. We implement our framework in PyTorch [46], using the geometric-deep-learning library PyTorch Geometric [47].

## 4.1 Shape Classification

The goal of shape classification is to assign each input mesh to a category (*e.g.*, chair, table). For the experiments on this task, we use a simple architecture consisting of a two stacked residual blocks, each containing two *primal-dual convolutional layers* and each attached to a pooling layer at its output. Similarly to [12], we insert a global average pooling layer after the last pooling layer, and we apply it on the features of the dual graph. The output of the average pooling layer is processed by a two-layer perceptron with ReLU activation, which predicts the class label for the input mesh. The network is trained using cross-entropy on the predicted labels. We do not perform any form of data augmentation. Additional details on the architecture parameters may be found in the Supplementary Material.

**SHREC dataset.** We use the lower-resolution version of the SHREC dataset [48] provided by [12], which consists of watertight meshes with 750 edges and 500 faces each. The dataset is made up of 600 samples from 30 different classes, with each class containing 20 samples.

Similarly to [12], we perform the evaluation on two types of dataset splits: *split 16* – where for each class 16 samples are used for training and 4 for testing – and *split 10* – in which the samples of each class are subdivided equally between training and the test set.

Following the same setup as [12], we limit the number of training epochs to 200, and for both *split 16* and *split 10* we randomly generate 3 sets and average our results over them. Table 1 shows that our method outperforms [12] by 4.6% on average over the splits. In addition, we also outperform the other baselines, which are volumetric and based on geometry images, by up to 36.5%.

| Method | Split 16 | Split 10 |
|---|---|---|
| Ours | **99.7%** | **99.1%** |
| MeshCNN [12] | 98.6% | 91.0% |
| GWCNN [49] | 96.6% | 90.3% |
| GI [50] | 96.6% | 88.6% |
| SN [51] | 48.4% | 52.7% |
| SG [52] | 70.8% | 62.6% |

Table 1: Classification accuracy on test set, SHREC dataset (comparisons from [12]).

**Cube Engraving dataset.** A second mesh-classification experiment is conducted on Cube Engraving dataset released by [12], which was generated using samples from the *MPEG-7* binary shape [53] dataset and insetting them into cubes with random position and orientation. Therefore, achieving good performance on this dataset requires learning distinctive features of the inset objects while neglecting the uninformative faces of the cubes.

The dataset consists of objects engraved in cubes and distributed in 22 classes with 200 samples per class (170 training samples and 30 test samples). Table 2 shows the results of this evaluation, in which our approach improves the accuracy by 2.23% with respect to the baseline of Hanocka et al. [12] and by 30.13% with respect to the point-cloud based PointNet++ [54].

| Method | Test accuracy |
|---|---|
| Ours | **94.39%** |
| MeshCNN [12] | 92.16% |
| PointNet++ [54] | 64.26% |

Table 2: Classification accuracy on test set, Cube Engraving dataset (comparisons from [12]).

## 4.2 Shape Segmentation

We evaluate our approach also on the task of shape segmentation, which consists in predicting a class label for each element (face, vertex, or edge) of a given mesh. In particular, since our method naturally aggregates information on clusters of faces, we predict a class label for each mesh face. Similarly to [12], we use a U-Net [55] encoder-decoder architecture with skip connections. The encoder is formed of 3 residual blocks with convolution layers, each followed by pooling. The decoder consists of 3 unpooling layers, each followed by a convolutional layer. The output of the network is a pair

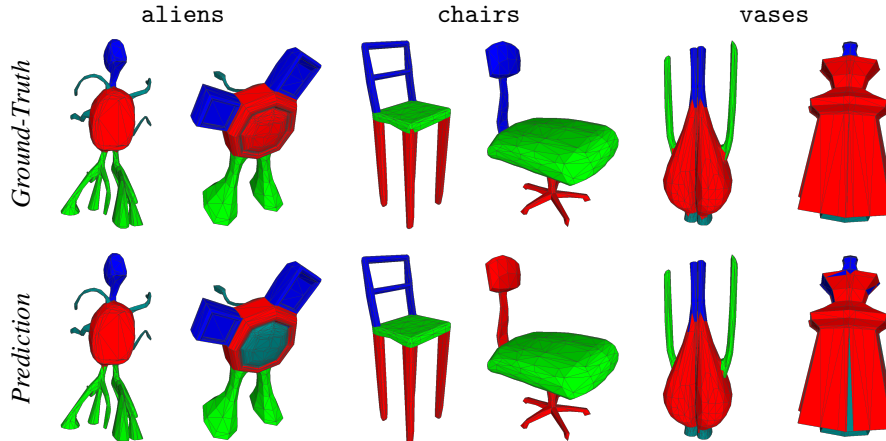

Figure 7: Example segmentations from the COSEG dataset. Same color corresponds to same class label. Due to its ability to create hierarchical clusters of faces, our approach predicts very accurate segmentations. The main failure case of our method consists in associating clusters to the wrong category.

of primal/dual graphs in the original input resolution. Every node of the resulting primal graph (uniquely associated to one of the input mesh faces) is associated with a per-class score, which is trained using cross-entropy loss for 1000 epochs. We perform experiments on two benchmarks for the task: COSEG [56] and Human Body [57]. Since these datasets contain ground-truth labels on the mesh edges, we convert the edge annotations into per-face labels by majority voting, selecting for each face the class label that is assigned to most edges in the face.

**Metrics.** Since our method and [12] are trained with labels on different mesh elements (faces and edges, respectively) performing a fair comparison for the shape segmentation task is challenging. As our method is trained on mesh faces, we measure performance according to the percentage of correctly-labelled mesh faces (*Face labels*). In order to compare with [12], we convert the edge labels predicted by the latter into face labels using the same procedure used to generate ground-truth, based on majority-voting. To ensure a fair comparison, in the very few cases (approximately $0.3\% \sim 0.7\%$ of the total) in which [12] predicts 3 different labels for the edges of a face, we do not consider the face in the evaluation. For reference, we also report the classification accuracy obtained by Hanocka et al. [12] on edge labels, which the method is trained on (*Edge labels*). This accuracy requires predicting a single label for each mesh edge, therefore a direct comparison with our method on this metric is not possible, since this would require converting per-face labels to per-edge single labels. However, we provide a more extensive analysis of the metrics in Sec. H of the Supplementary Material, where we perform comparisons also according to edge-based metrics and show that our method outperforms [12] also on these types of accuracies. We rerun each of the experiments of [12] 4 times – using the official code and parameters provided – and we report the best accuracy obtained.

**COSEG Dataset.** For evaluation we use the same dataset splits as [12]. The input meshes are divided into the following three categories, for each of which a different experiment is performed:

- Category `aliens`, consisting of 198 samples (169 training set, 29 test set) with 4 class labels;
- Category `chairs`, that contains 397 samples (337 training set, 60 test set) with 3 class labels;
- Category `vases`, made of 297 samples (252 training set, 45 test set) with 4 class labels.

Table 3 compares the results obtained by our method and by [12] on the three categories in the COSEG dataset, using the metrics defined above. For the metric on face labels our method outperforms [12] by up to $4.24\%$. We believe this performance boost to be motivated by the way our approach aggregates information across faces, which allows to identify structurally coherent clusters in a mesh more naturally than methods based on collapsing mesh edges. This is qualitatively illustrated in Fig. 7, where we show that our method produces high-quality segmentation masks. Fig. 8 further shows that the cluster of faces formed through the attention-driven pooling operation indeed tend to abstract to larger areas which carry common semantic information; we stress that the network does not always find such structures, but also highlight that no supervision on the face clusters was provided during training.

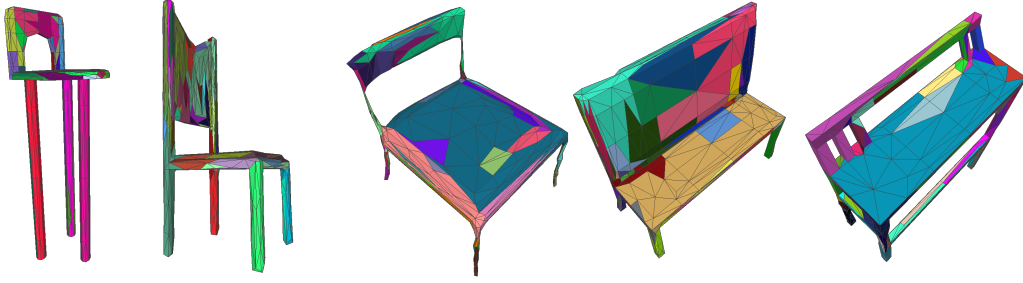

Figure 8: Example of clusters internally formed by PD-MeshNet on samples from the COSEG `chair` dataset; same color corresponds to same cluster. The clusters shown are outputted by the last pooling layer in the network (cf. Supplementary Material for further details on the architecture). On the left, the legs of the chairs are consistently identified as clusters; on the right, larger common planar structures are found across different samples.

| Category | Method | Metric | |
| | | Edge labels | Face labels |
| --- | --- | --- | --- |
| aliens | MeshCNN [12] | 95.35% | 96.26% |
| | Ours | - | **98.18%** |
| chairs | MeshCNN [12] | 92.65% | 92.99% |
| | Ours | - | **97.23%** |
| vases | MeshCNN [12] | 91.96% | 92.38% |
| | Ours | - | **95.36%** |

Table 3: Test accuracy on mesh segmentation task, COSEG dataset.

| Method | Metric | |
| | Edge labels | Face labels |
| --- | --- | --- |
| MeshCNN [12] | 84.05% | 85.39% |
| Ours | - | **85.61%** |

Table 4: Test accuracy on mesh segmentation task, Human Body dataset.

**Human Body Dataset.** The dataset consists of 381 training samples and 18 test samples, both with a resolution of 1500 faces. Similarly to Hanocka et al. [12], we generate 20 augmented versions of each training sample by randomly shifting the vertices along the edges. As shown in Table 4, also in this dataset our approach outperforms the baseline. However, the performance gap is lower than for the other experiment.

## 5   Conclusions

In this paper, we presented PD-MeshNet, a novel deep-learning framework for processing 3D meshes. Our approach combines an attention-based convolution operation on two graphs constructed from an input 3D mesh with a task-driven pooling operation that corresponds to clustering mesh faces. We achieve a performance superior or comparable to the state-of-the-art on the tasks of shape segmentation and shape classification. Our method is the first to create a connection between graph-based and ad-hoc methods for mesh processing. We believe that further developing this connection is an interesting avenue for future work.

We empirically noticed a performance drop when the network gets too deep (due to the larger number of parameters associated with the two graphs and the more complex optimization landscape), a problem shared by graph neural networks in general [58, 59]. Moreover, pooling layers can be sensitive to their threshold parameters, with too-aggressive a pooling causing a degradation of results. Addressing these limitations is an interesting direction for future work.

Finally, we believe that the higher-level abstraction suggested by our pooling operation could set the basis for a hierarchical representation of objects and, subsequently, scenes. Such representation would be beneficial to all applications requiring a high-level semantic understanding of the environment.

## Broader Impact

Processing of 3D data, in the form of point-cloud, voxel-grids, or meshes finds important applications in several fields, including computer graphics, vision, and robotics. Further developments of this work, which proposes a novel framework for mesh processing based on deep leaning, could have a benefit on several real-world applications, including augmented and virtual reality, robotics, and spatial 3D scene understanding of environments. These applications can have profound positive implications for the future of our society, by, for example, improving the quality of virtual social interactions, or increasing the spatial-awareness of current robotic systems to integrate them in our everyday life.

## Acknowledgments and Disclosure of Funding

This work was partially funded by ARL DCIST CRA W911NF-17-2-0181, and MIT Lincoln Laboratory "Task Execution with Semantic Segmentation" program. This work was also partially supported by the National Centre of Competence in Research (NCCR) Robotics through the Swiss National Science Foundation (SNSF) and the SNSF-ERC Starting Grant. Francesco Milano was partially supported by the Zeno Karl Schindler Foundation Master Thesis Grant.

## Footnotes

[2]We recall that a triangle mesh is 2-manifold if its surface is everywhere locally homeomorphic to a disk; in particular, an edge is *non-manifold* if it has more than two incident triangles [6, pp. 11-12].

[3] Equivalently, $K$ can be expressed as a fraction of the number of primal nodes in the input graph.

[4] It should be noted that after a pooling operation, the dual graph can no longer be interpreted as the medial graph of the $\mathcal{G}(\mathcal{M})$. Indeed, the 4-regularity property does not hold anymore (cf. Fig. 5).

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
