[Supplementary Material]

# Supplementary Material of Primal-Dual Mesh Convolutional Neural Networks

**Francesco Milano**[*]
ETH Zurich, Switzerland
`fmilano@student.ethz.ch`

**Antonio Loquercio**
Robotics and Perception Group
University of Zurich, Switzerland
`loquercio@ifi.uzh.ch`

**Antoni Rosinol**
SPARK Lab
MIT, USA
`arosinol@mit.edu`

**Davide Scaramuzza**
Robotics and Perception Group
University of Zurich, Switzerland
`sdavide@ifi.uzh.ch`

**Luca Carlone**
SPARK Lab
MIT, USA
`lcarlone@mit.edu`

## Contents

[*]Work performed while at MIT.

# A  Classification of related work

Table A.1 summarizes the most relevant related works that perform learning-based processing of meshes. Each method is classified according to the type of approach (graph-based/non-graph-based), to the type of pooling (no pooling, mesh-based/non-mesh-based, task-driven/non-task-driven), to the implementation of a form of dynamic feature aggregation, and to the type of task for which the method is designed.

| Method | Type of method | | | Pooling | | | Dynamic aggr. | Task |
|---|---|---|---|---|---|---|---|---|
| | Graph-based | | Non-graph | Non-mesh based | Mesh-based | | | |
| | Spatial | Spectral | | | Non-task-driven | Task-driven | | |
| GCNN [1] | ✗ | | | | | | | Shape correspondence |
| ACNN [2] | ✗ | | | | | | | Shape correspondence/description/retrieval |
| MoNet [3] | ✗ | | | | | | | Shape correspondence |
| FeaStNet [4] | ✗ | | | ✗ ([5]) | | | ✗ | Shape correspondence/segmentation |
| GEM-CNN [6] | ✗ | | | | | | | Shape correspondence/classification |
| DCM-Net [7] | ✗ ([8]) | | | | ✗ ([9]/[10]) | | ✗ | Scene segmentation |
| SN [11] | | ✗ | | | | | | Deformation prediction/generative model |
| CoMA [12] | | ✗ | | | ✗ ([9]) | | | Generative model |
| TangentConv [13] | | | ✗ | ✗ (grid-based) | | | | Scene segmentation |
| TextureNet [14] | | | ✗ | ✗ (sample-based) | | | | Scene segmentation |
| MeshCNN [15] | | | ✗ | | | ✗ | | Shape classification/segmentation |
| MeshNet [16] | | | ✗ | | | | | Shape classification/retrieval |
| SyncSpecCNN [17] | | ✗ | | ✗ (spectral) | | | | Shape segmentation/keypoint prediction |
| SpiralNet++ [18] | ✗ | | | | ✗ ([9]) | | ✗ | Shape correspondence/classification/reconstruction |
| **PD-MeshNet** | ✗ | | | | | ✗ | ✗ | Shape classification/segmentation |

Table A.1: Classification of most relevant related work that perform learning-based processing of meshes.

## A.1  Forms of dynamic aggregation

For each node in an input graph, FeaStNet [4] dynamically learns the correspondence between its neighboring nodes and the filter weights based on the features of the node and of its neighbors. DCM-Net [7] dynamically determines the neighbors of each node in its convolution based on Euclidean distance.

# B  Analogy between line graph and medial graph

In the following, we show that, for an edge-manifold, triangle mesh $\mathcal{M}$, the medial graph of the mesh graph $\mathcal{G}(\mathcal{M})$ coincides with the line graph of the primal graph, *i.e.*,, $M(\mathcal{G}(\mathcal{M})) \equiv \mathcal{L}(\mathcal{P}(\mathcal{M}))$ (cf. Theorem 1). For simplicity, we further assume the mesh to be of genus 0; a similar result can be obtained for embeddings on surfaces of higher genus by extending Lemma 2. For convenience, we report here the definitions of *line graph* and *medial graph*:

- The *line graph* $\mathcal{L}(G)$ of a graph $G$ has a node for each edge of $G$, and two nodes in $\mathcal{L}(G)$ are adjacent if and only if the corresponding edges in $G$ have a node in common [19, p. 20].
- The *medial graph* $M(G)$ of a *plane* graph $G$ - or more generally of a graph embedded on a higher-genus surface [19, p. 723] - has a node for each edge of $G$ and an edge between two nodes if the edges of $G$ corresponding to the two nodes are adjacent on one face of $G$ [20].

**Assumption 1.** *$\mathcal{M}$ is an edge-manifold, triangle mesh of genus 0.*

**Lemma 1.** *Under Assumption 1, $\mathcal{P}(\mathcal{M})$ is a cubic (*i.e.,*, 3-regular), planar graph.*

*Proof.*  3-regularity follows by definition of simplex mesh, together with the fact that $\mathcal{M}$ is triangular by hypothesis. The further hypothesis of edge-manifoldness yields the fact that $\mathcal{P}(\mathcal{M})$ is a planar graph, by definition of edge-manifoldness. □

**Lemma 2** (Ore [21])**.** *The medial graph of a cubic plane graph coincides with its line graph.*

**Corollary 1.** $M(\mathcal{P}(\mathcal{M})) \equiv \mathcal{L}(\mathcal{P}(\mathcal{M}))$.

*Proof.*  The proposition follows directly from Lemmas 1 and 2. □

**Lemma 3** (Gross and Yellen [19, p. 724])**.** *The medial graph of a plane graph – and more generally of a graph embedded on a surface – is identical to the medial graph of its dual graph.*

**Theorem 1.** $M(\mathcal{G}(\mathcal{M})) \equiv \mathcal{L}(\mathcal{P}(\mathcal{M}))$.

*Proof.* By definition, our primal graph $\mathcal{P}(\mathcal{M})$ is the (topologically) dual graph of $\mathcal{G}(\mathcal{M})$ [22], hence $M(\mathcal{G}(\mathcal{M})) \equiv M(\mathcal{P}(\mathcal{M}))$ from Lemma 3. From Corollary 1 it also holds that $M(\mathcal{P}(\mathcal{M})) \equiv \mathcal{L}(\mathcal{P}(\mathcal{M}))$; thus, $M(\mathcal{G}(\mathcal{M})) \equiv \mathcal{L}(\mathcal{P}(\mathcal{M}))$. □

## C   Further details on graphs

### C.1   Handling non-manifoldness

In the following, we show that PD-MeshNet can be easily extended to handle non-manifold meshes. The definition of the dual features that we borrow from [15] relies on the assumption that each edge of the mesh to which a feature is assigned is shared by exactly two faces, hence implying edge-manifoldness. However, the assumption of edge-manifoldness of the input mesh is not required by PD-MeshNet for the convolution operation – which can handle an arbitrary number of neighbors – nor is necessary for the pooling operation – which does not alter the topological type of the mesh, as opposed, *e.g.*, to the *edge collapse* used in [15] (cf. [23] and Fig. C.1). Therefore, by simply adding nodes in the dual graph or changing the definition of features, PD-MeshNet can handle non-manifold edges. Considering the non-manifold edge shared by faces $A$, $B$ and $C$ in Fig. C.2, we identify the following two possible solutions:

1. Two separate dual nodes $\{A, B\}$ and $\{A, C\}$ can be defined; since each is associated to exactly two faces (respectively $A, B$ and $A, C$), the same features as [15] (cf. Sec. 3.1 in the main paper) can be used.

2. A single dual node $\{A, B, C\}$ can be inserted in the graph. In this case, different features need to be defined; one possibility is to concatenate – or average – the features that can be defined between faces $A, B$ and faces $A, C$ (cf. Sec. 3.1 in the main paper).

(a) Since the vertex $c$ on the left side has valence 3, collapsing the edge $ab$ causes the formation of a non-manifold edge (in yellow on the right side)

(b) Collapsing the edge $ab$ on the left side of the figure causes the vertex $c$ to become of valence 3 (cf. right side)

Figure C.1: The *edge collapse* operation used for pooling in [15] cannot collapse edges adjacent to valence-3 vertices, because it would break the assumption of edge-manifoldness required by the convolution operation of [15]; a valence-3 vertex is formed whenever an edge adjacent to a valence-4 vertex is collapsed. On the contrary, PD-MeshNet does not require edge-manifoldness to define its convolution operation, and its pooling operation does not alter the topological type of the mesh.

Figure C.2: Shown in red is an example of non-manifold edge shared by three faces $A, B,$ and $C$.

## C.2 Dual-graph features

We provide in the following a detail about the definition of the dual features. For each dual node $\{A, B\}$ we replace the internal angles $\gamma_A$ and $\gamma_B$ used as features by [15] with the *edge-to-edge ratios* of the faces $A$ and $B$, *i.e.*, with reference to Fig. 2 in the main paper, $\frac{\|ab\|}{\|ad\|}$, $\frac{\|ab\|}{\|bd\|}$ and $\frac{\|ab\|}{\|bc\|}$, $\frac{\|ab\|}{\|ac\|}$ are used respectively in place of $\gamma_A$ and $\gamma_B$. It should be noted that the two types of features encode the same type of information. Indeed, for instance by the law of cosines one has:

$$
\begin{aligned}
2\|ad\|\|bd\|\cos(\gamma_A) &= \|ad\|^2 + \|bd\|^2 - \|ab\|^2 \Rightarrow \\
\Rightarrow 2\cos(\gamma_A) &= \frac{\|ad\|}{\|bd\|} + \frac{\|bd\|}{\|ad\|} - \frac{\|ab\|}{\|ad\|}\frac{\|ab\|}{\|bd\|} \Rightarrow \\
\Rightarrow \cos(\gamma_A) &= \frac{1}{2}\left(\frac{k_1}{k_2} + \frac{k_2}{k_1} - k_1 k_2\right),
\end{aligned}
\tag{1}
$$

with $k_1 \coloneqq \frac{\|ab\|}{\|ad\|}$ and $k_2 \coloneqq \frac{\|ab\|}{\|bd\|}$.

## C.3 Dual-graph configurations

The following section introduces different possible configurations of the dual graph; the ablation study in Sec. C.4 shows that the choice of any of these configurations produces no significant differences in performance.

We define three admissible configurations of the dual graph, depending on whether a single or a double dual node is defined for each edge of the mesh, and on the directness of the edges of the dual graph:

**Single dual nodes.** For each pair of adjacent mesh faces $A, B \in \mathcal{F}(\mathcal{M})$, a *single* node $\{A, B\}$ is inserted in the graph. This is the configuration introduced in the main paper, and we refer to it as *dual-graph configuration* Ⓐ. As previously mentioned, this is the configuration implicitly used by the method of Hanocka et al. [15], and for each dual node we therefore define the same features as [15], up to the implementation detail mentioned in Sec. C.2. In symbols, with reference to Fig. 2 in the main paper, one has the following feature vector for a generic dual node $\{A, B\}$:

$$
\tilde{\boldsymbol{f}}_{\{A,B\}} = \left[\theta_{AB}, \frac{\|ab\|}{h_A}, \frac{\|ab\|}{h_B}, \frac{\|ab\|}{\|ad\|}, \frac{\|ab\|}{\|bd\|}, \frac{\|ab\|}{\|bc\|}, \frac{\|ab\|}{\|ac\|}\right]^{\mathsf{T}}.
\tag{2}
$$

It should be noted that the both the edge-to-height ratios and the edge-to-edge ratios in (2) are defined up to rotational symmetry, *i.e.*, one can have:

$$
\tilde{\boldsymbol{f}}_{\{A,B\}} = \left[\theta_{AB}, \frac{\|ab\|}{h_B}, \frac{\|ab\|}{h_A}, \frac{\|ab\|}{\|bc\|}, \frac{\|ab\|}{\|ac\|}, \frac{\|ab\|}{\|ad\|}, \frac{\|ab\|}{\|bd\|}\right]^{\mathsf{T}}
\tag{3}
$$

in alternative to (2). To solve the ordering ambiguity, similarly to [15] we further sort both the edge-to-height ratios and the edge-to-edge ratios in increasing order. Each edge in the graph is *undirected*, *i.e.*,, each node $\{A, B\}$ is connected to its neighboring nodes $\{A, M\}, M \in \mathcal{N}_A\backslash\{B\}$ and $\{B, N\}, N \in \mathcal{N}_B\backslash\{A\}$ both with *incoming* (e.g., $\{A, M\} \to \{A, B\}$) and *outgoing* (e.g., $\{A, B\} \to \{A, M\}$) edges (Fig. C.3a).

**Double dual nodes.** Alternatively, one can map each pair of adjacent faces $A, B \in \mathcal{F}(\mathcal{M})$ to a *double* dual node, by inserting a node $A \to B$ and a node $B \to A$ in the dual graph. This allows avoiding the symmetry ambiguity in the features: without loss of generality, we assign to the dual node $A \to B$ the subset of the features in (2) that represent the geometry of face $A$ *as seen from face $B$*, and to node $B \to A$ the features that represent the geometry of face $B$ *as seen from face $A$*, *i.e.*,, with reference to Fig. 2 in the main paper:

$$
\tilde{\boldsymbol{f}}_{A \to B} = \left[\theta_{AB}, \frac{\|ab\|}{h_A}, \frac{\|ab\|}{\|ad\|}, \frac{\|ab\|}{\|bd\|}\right]^{\mathsf{T}}, \qquad \tilde{\boldsymbol{f}}_{B \to A} = \left[\theta_{AB}, \frac{\|ab\|}{h_B}, \frac{\|ab\|}{\|bc\|}, \frac{\|ab\|}{\|ac\|}\right]^{\mathsf{T}}.
\tag{4}
$$

The edges in the graph can be both *undirected* and *directed*, *i.e.*,, a generic dual node $A \to B$ can be connected to the neighboring nodes $M \to A, M \in \mathcal{N}_A\backslash\{B\}$ and $B \to N, N \in \mathcal{N}_B\backslash\{A\}$:

- Both with an *incoming* and an *outgoing* edge, as done for generic graphs in [24] (where the neighboring nodes are instead of the form $A \to M$ and $N \to B$). We refer to this configuration as *dual-graph configuration* Ⓑ (Fig. C.3b);

(a) Configuration Ⓐ

(b) Configuration Ⓑ

(c) Configuration Ⓒ

Figure C.3: Admissible dual-graph configurations for an example mesh.

- Only with edges outgoing from $M \to A$ and incoming in $B \to N$, *i.e.,*, of the form $(M \to A) \to (A \to B)$ and $(A \to B) \to (B \to N)$. We term this configuration *dual-graph configuration* Ⓒ (Fig. C.3c).

## C.4 Ablation study

We experimentally noticed no substantial differences in performance across the three configurations, and in the main paper we therefore reported for simplicity the results obtained for dual-graph configuration Ⓐ. Below we present the results of an ablation study performed on the SHREC dataset for the mesh classification task and on the Human Body dataset for the mesh segmentation task, using for both the parameters provided in Sec. F. The training process is run for 200 epochs for the classification experiment and for 20 epochs (considering the whole augmented dataset) for the segmentation experiment. As shown in Tables C.2 and C.3, in the mesh segmentation task the single-node configuration (Ⓐ) performs slightly better than the double-node configurations (Ⓑ and Ⓒ), while the latter outperform configuration Ⓐ by a small margin in the mesh classification task.

| Dual-graph configuration | Split 16 | Split 10 |
|---|---|---|
| Ⓐ | 99.72% | 99.11% |
| Ⓑ | **100.00%** | **99.78%** |
| Ⓒ | **100.00%** | 99.67% |

Table C.2: Classification accuracy on the SHREC dataset for the three admissible dual-graph configurations. Similarly to what done in the main paper, we limit the training to 200 epochs, and for each split we randomly generate 3 sets and average the results over these.

| Dual-graph configuration | Face labels |
|---|---|
| Ⓐ | **84.78%** |
| Ⓑ | 84.18% |
| Ⓒ | 83.53% |

Table C.3: Segmentation accuracy on the Human Body dataset for the three admissible dual-graph configurations. Similarly to what done in the main paper, we generate 20 augmented versions of each training sample by randomly sliding vertices along the mesh edges. Mean pooling aggregation is used.

## D  Further details on convolution

In the following section, we provide the mathematical details of the convolution operation used by PD-MeshNet according to the different dual-graph configurations. We use the notation introduced in the main paper, and we further define the following symbols:

$\xi, \tilde{\xi}$ — Non-linear activation functions (ReLU) associated with the primal and dual layer respectively

$\boldsymbol{W}, \tilde{\boldsymbol{W}}$ — Shared learnable kernel used to multiply respectively primal- and dual- node features

$\eta, \tilde{\eta}$ — Non-linear activation functions (Leaky-ReLU) used before softmax when computing primal- and dual- attention coefficients respectively

$\boldsymbol{a}, \tilde{\boldsymbol{a}}$ — Learnable attention parameters used in the computation of the primal- and dual-attention coefficients respectively

$\boldsymbol{f_A}||\boldsymbol{f_B}$ — Vertical concatenation between features $\boldsymbol{f_A}$ and $\boldsymbol{f_B}$

### D.1  Dual convolution

In the following, we assume $(A, B)$ to be a pair of adjacent mesh faces. It should be noted that for each dual node the neighborhoods that index the summations in the equations in the section below match exactly those defined in the medial graph $M(\mathcal{G}(\mathcal{M}))$ (cf. Fig. C.3 and Fig. 1c in the main paper); optionally, self-loops can be inserted in the graph, and multiple attention heads can be used, with the resulting features being either concatenated or averaged.

### D.1.1  Dual-graph configuration Ⓐ

For the generic dual node $\{A, B\}$, the layer outputs the following feature:

$$\tilde{\boldsymbol{f}}'_{\{A,B\}} = \tilde{\xi} \left( \sum_{M \in \mathcal{N}_A \setminus \{B\}} \tilde{\alpha}_{\{A,M\},\{A,B\}} \tilde{\boldsymbol{f}}_{\{A,M\}} \tilde{\boldsymbol{W}} + \sum_{N \in \mathcal{N}_B \setminus \{A\}} \tilde{\alpha}_{\{B,N\},\{A,B\}} \tilde{\boldsymbol{f}}_{\{B,N\}} \tilde{\boldsymbol{W}} \right). \quad (5)$$

The attention coefficient $\tilde{\alpha}_{\{A,M\},\{A,B\}}$ defined on the generic dual edge $\{A,M\} \to \{A,B\}$, with $M \in \mathcal{N}_A\backslash\{B\}$, can be found as follows:

$$\tilde{\alpha}_{\{A,M\},\{A,B\}} = \frac{e^{\tilde{\eta}\left(\tilde{a}^T\left[\tilde{f}_{\{A,M\}}\tilde{W}\|\tilde{f}_{\{A,B\}}\tilde{W}\right]\right)}}{\displaystyle\sum_{K\in\mathcal{N}_A\backslash\{B\}} e^{\tilde{\eta}\left(\tilde{a}^T\left[\tilde{f}_{\{A,K\}}\tilde{W}\|\tilde{f}_{\{A,B\}}\tilde{W}\right]\right)} + \sum_{N\in\mathcal{N}_B\backslash\{A\}} e^{\tilde{\eta}\left(\tilde{a}^T\left[\tilde{f}_{\{B,N\}}\tilde{W}\|\tilde{f}_{\{A,B\}}\tilde{W}\right]\right)}}.$$

(6)

Similarly, the attention coefficient $\tilde{\alpha}_{\{B,N\},\{A,B\}}$ defined on the generic dual edge $\{B,N\} \to \{A,B\}$, with $N \in \mathcal{N}_B\backslash\{A\}$, can be computed as:

$$\tilde{\alpha}_{\{B,N\},\{A,B\}} = \frac{e^{\tilde{\eta}\left(\tilde{a}^T\left[\tilde{f}_{\{B,N\}}\tilde{W}\|\tilde{f}_{\{A,B\}}\tilde{W}\right]\right)}}{\displaystyle\sum_{M\in\mathcal{N}_A\backslash\{B\}} e^{\tilde{\eta}\left(\tilde{a}^T\left[\tilde{f}_{\{A,M\}}\tilde{W}\|\tilde{f}_{\{A,B\}}\tilde{W}\right]\right)} + \sum_{K\in\mathcal{N}_B\backslash\{A\}} e^{\tilde{\eta}\left(\tilde{a}^T\left[\tilde{f}_{\{B,K\}}\tilde{W}\|\tilde{f}_{\{A,B\}}\tilde{W}\right]\right)}}.$$

(7)

### D.1.2 Dual-graph configuration Ⓑ

For the generic dual node $A \to B$, the layer outputs the following feature:

$$\tilde{f}'_{A\to B} = \tilde{\xi}\left(\sum_{M\in\mathcal{N}_A\backslash\{B\}} \tilde{\alpha}_{M\to A,A\to B}\tilde{f}_{M\to A}\tilde{W} + \sum_{N\in\mathcal{N}_B\backslash\{A\}} \tilde{\alpha}_{B\to N,A\to B}\tilde{f}_{B\to N}\tilde{W}\right). \quad (8)$$

The attention coefficient $\tilde{\alpha}_{M\to A,A\to B}$ defined on the generic dual edge $(M \to A) \to (A \to B)$, with $M \in \mathcal{N}_A\backslash\{B\}$, can be found as follows:

$$\tilde{\alpha}_{M\to A,A\to B} = \frac{e^{\tilde{\eta}\left(\tilde{a}^T\left[\tilde{f}_{M\to A}\tilde{W}\|\tilde{f}_{A\to B}\tilde{W}\right]\right)}}{\displaystyle\sum_{K\in\mathcal{N}_A\backslash\{B\}} e^{\tilde{\eta}\left(\tilde{a}^T\left[\tilde{f}_{K\to A}\tilde{W}\|\tilde{f}_{A\to B}\tilde{W}\right]\right)} + \sum_{N\in\mathcal{N}_B\backslash\{A\}} e^{\tilde{\eta}\left(\tilde{a}^T\left[\tilde{f}_{B\to N}\tilde{W}\|\tilde{f}_{A\to B}\tilde{W}\right]\right)}}.$$

(9)

Similarly, the attention coefficient $\tilde{\alpha}_{B\to N,A\to B}$ defined on the generic dual edge $(B \to N) \to (A \to B)$, with $N \in \mathcal{N}_B\backslash\{A\}$, can be computed as:

$$\tilde{\alpha}_{B\to N,A\to B} = \frac{e^{\tilde{\eta}\left(\tilde{a}^T\left[\tilde{f}_{B\to N}\tilde{W}\|\tilde{f}_{A\to B}\tilde{W}\right]\right)}}{\displaystyle\sum_{M\in\mathcal{N}_A\backslash\{B\}} e^{\tilde{\eta}\left(\tilde{a}^T\left[\tilde{f}_{M\to A}\tilde{W}\|\tilde{f}_{A\to B}\tilde{W}\right]\right)} + \sum_{K\in\mathcal{N}_B\backslash\{A\}} e^{\tilde{\eta}\left(\tilde{a}^T\left[\tilde{f}_{B\to K}\tilde{W}\|\tilde{f}_{A\to B}\tilde{W}\right]\right)}}.$$

(10)

### D.1.3 Dual-graph configuration Ⓒ

For the generic dual node $A \to B$, the layer outputs the following feature:

$$\tilde{f}'_{A\to B} = \tilde{\xi}\left(\sum_{M\in\mathcal{N}_A\backslash\{B\}} \tilde{\alpha}_{M\to A,A\to B}\tilde{f}_{M\to A}\tilde{W}\right). \quad (11)$$

The attention coefficient $\tilde{\alpha}_{M\to A,A\to B}$ defined on the generic dual edge $(M \to A) \to (A \to B)$, with $M \in \mathcal{N}_A\backslash\{B\}$, can be found as follows:

$$\tilde{\alpha}_{M\to A,A\to B} = \frac{e^{\tilde{\eta}\left(\tilde{a}^T\left[\tilde{f}_{M\to A}\tilde{W}\|\tilde{f}_{A\to B}\tilde{W}\right]\right)}}{\displaystyle\sum_{K\in\mathcal{N}_A\backslash\{B\}} e^{\tilde{\eta}\left(\tilde{a}^T\left[\tilde{f}_{K\to A}\tilde{W}\|\tilde{f}_{A\to B}\tilde{W}\right]\right)}}. \quad (12)$$

### D.2 Primal convolution

For all dual-graph configurations, the output feature of a generic primal node $A$ can simply be found as:

$$f'_A = \xi\left(\sum_{M\in\mathcal{N}_A} \alpha_{M,A}f_M W\right). \quad (13)$$

What varies across the different dual-graph configurations is the attention coefficient $\alpha_{M,A}$ associated to each generic primal edge $M \to A$, with $M \in \mathcal{N}_A$, as we detail below.

**Dual-graph configuration Ⓐ.**

$$\alpha_{M,A} = \frac{e^{\eta\left(\boldsymbol{a}^T \tilde{\boldsymbol{f}}'_{\{A,M\}}\right)}}{\displaystyle\sum_{B \in \mathcal{N}_A} e^{\eta\left(\boldsymbol{a}^T \tilde{\boldsymbol{f}}'_{\{A,B\}}\right)}}. \tag{14}$$

**Dual-graph configuration Ⓑ and Ⓒ.**

$$\alpha_{M,A} = \frac{e^{\eta\left(\boldsymbol{a}^T \tilde{\boldsymbol{f}}'_{M \to A}\right)}}{\displaystyle\sum_{B \in \mathcal{N}_A} e^{\eta\left(\boldsymbol{a}^T \tilde{\boldsymbol{f}}'_{B \to A}\right)}}. \tag{15}$$

# E  Further details on pooling

## E.1  Implementation details

As shown in Sec. 3.3, contracting edges in the primal graph according to the quantity (1) in the main paper causes the formation of clusters of faces that belong in general to a triangle fan. One special case that can occur is the one in which a single primal edge not selected for contraction according to (1) from the main paper prevents the formation of a *closed* triangle fan. Consider the example of Fig. E.4: the edges between the pairs of faces $(A, B)$, $(B, E)$, $(C, D)$, and $(D, E)$ are selected for contraction according to (1) from the main paper, but the one before faces $A$ and $C$ is not. As a consequence, one would have the formation of a new primal node (cluster of faces) $ABCDE$ with a self-loop corresponding to the edge originally between primal nodes $A$ and $C$. To avoid generating such self-loop, we force the single edge that prevents the formation of a *closed* triangle fan (between nodes $A$ and $C$ in the example) to also be collapsed.

Figure E.4: Example of contraction of a primal edge performed even though the quantity (1) from the main paper is not among the largest $K$. The edges selected for contraction according to (1) from the main paper are shown in green on the left side. The primal edge between faces $A$ and $C$ is also contracted because it is the only one that prevents the faces $A, B, C, D$ and $E$ from forming a *closed* triangle fan after pooling.

# F  Architectures and training details

## F.1  Classification

The results reported for the classification experiments are obtained using the architecture shown in Fig. F.5, with input graphs of dual-graph configuration Ⓐ, and using 3 attention heads with concatenation of the output features across the different heads. Each residual convolutional block contains two stacked convolutional layers with a single skip connection and each followed by group normalization (GN) [25] and ReLU activation. The network is trained for 200 epochs using cross-entropy loss and a fixed learning rate of $2e{-}4$. A batch size of 16 is used.

A summary of the architecture parameters can be found in Tab. F.4.

Figure F.5: Classification architecture. The graphs constructed from an example input mesh (from the SHREC dataset [26]) are passed through 2 stacked residual convolutional blocks each followed by a pooling layer; average pooling is then applied on the output of the last pooling layer, followed by two fully-connected layers that predict the class of the input mesh.

Figure F.6: U-Net architecture used for segmentation. The graphs constructed from an example input mesh (from the COSEG dataset [27]) are passed through an encoder, which consists of 3 stacked residual convolutional blocks each followed by a pooling layer; the decoder consists of unpooling layers mirroring the pooling layers – each with skip connections from the encoder and preceded and followed by a convolutional layer – and brings the graphs to their original resolution. A final convolutional layer predicts a class label on each face of the input mesh (*i.e.*, on each primal node). The number in the top-right and bottom-right corners of each layer indicate the number of output primal and dual channels respectively, while those in the top-left and bottom-left corners represent the number of input primal/dual channels.

## F.2 Segmentation

Figure F.6 shows the U-Net architecture used for the segmentation experiments. Similarly to the classification experiment, the results reported in the main paper are obtained with dual-graph configuration Ⓐ, and each residual convolutional block contains two stacked convolutional layers with a single skip connection, both followed by group normalization and ReLU activation. The architecture is composed of an encoder with three levels of convolutional blocks each followed by a pooling layer; a single convolutional layer connects the encoder to the decoder, which is made of three levels that mirror the layers in the decoder. Each decoder level is made of a convolutional layer, an unpooling layer and a second convolutional layer. At the output of each unpooling layer, skip connections are inserted from the encoder to the decoder: the encoder features are averaged over

| Module type | #in channels (primal/dual) | #out channels (primal/dual) | Fract. primal edges pooled |
|---|---|---|---|
| Conv+GN+ReLU | $1 / 7$ | $64 * H / 64 * H$ | – |
| SC+Conv+GN+ReLU | $64 * H / 64 * H$ | $64 * H / 64 * H$ | – |
| Pooling | – | – | 0.2 |
| Conv+GN+ReLU | $64 * H / 64 * H$ | $128 * H / 128 * H$ | – |
| SC+Conv+GN+ReLU | $128 * H / 128 * H$ | $128 * H / 128 * H$ | – |
| Pooling | – | – | 0.2 |
| Avg pool | $128 * H / 128 * H$ | $- / 128 * H$ | – |
| Linear | $- / 128 * H$ | $- / 100$ | – |
| Linear | $- / 100$ | $- / C$ | – |

Table F.4: Parameters of the architecture used for the classification experiments (cf Fig. F.5). $H$ denotes the number of attention heads (3 in our experiments), while $C$ is the number of classes in the dataset (30 for the SHREC dataset, 22 for the Cube Engraving dataset). $SC$ indicates a skip connection from the previous module.

the attention heads and concatenated to the feature outputted by the unpooling layer. The second convolutional layer in each decoder level returns the output for the subsequent decoder level. A final convolutional layer predicts per-class labels for each class of the input mesh. We use 3 attention heads in each convolutional block of the encoder and 1 in each block of the decoder. We train the network for 1000 epochs with a learning rate of $1e-3$ using cross-entropy loss on the per-face class labels. We use a batch size of 16 for the COSEG experiments and a batch size of 12 for the experiments on the Human Body dataset.
A summary of the architecture parameters can be found in Tab. F.5.

| Level | Level type | Module type | #in channels (primal/dual) | #out channels (primal/dual) | Fract. primal edges pooled |
|---|---|---|---|---|---|
| 1 | Encoder | Conv+GN+ReLU | $1 / 7$ | $32 * H_e / 32 * H_e$ | – |
| 1 | Encoder | SC.+Conv+GN+ReLU | $32 * H_e / 32 * H_e$ | $32 * H_e / 32 * H_e$ | – |
| 1 | Encoder | Pooling | – | – | 0.3 |
| 2 | Encoder | Conv+GN+ReLU | $32 * H_e / 32 * H_e$ | $64 * H_e / 64 * H_e$ | – |
| 2 | Encoder | SC.+Conv+GN+ReLU | $64 * H_e / 64 * H_e$ | $64 * H_e / 64 * H_e$ | – |
| 2 | Encoder | Pooling | – | – | 0.3 |
| 3 | Encoder | Conv+GN+ReLU | $64 * H_e / 64 * H_e$ | $128 * H_e / 128 * H_e$ | – |
| 3 | Encoder | SC.+Conv+GN+ReLU | $128 * H_e / 128 * H_e$ | $128 * H_e / 128 * H_e$ | – |
| 3 | Encoder | Pooling | – | – | 0.3 |
| 4 | Single Conv | Conv+BN+ReLU | $128 * H_e / 128 * H_e$ | $256 * H_e / 256 * H_e$ | – |
| 4 | Single Conv | Average att. heads | $256 * H_e / 256 * H_e$ | $256 / 256$ | – |
| 3 | Decoder | Conv+BN+ReLU | $256 / 256$ | $128 / 128$ | – |
| 3 | Decoder | Unpooling | – | – | 0.3 (unpooling) |
| 3 | Decoder | SCE | $128 / 128$ | $128 * 2 / 128 * 2$ | – |
| 3 | Decoder | Conv+BN+ReLU | $256 / 256$ | $128 / 128$ | – |
| 2 | Decoder | Conv+BN+ReLU | $128 / 128$ | $64 / 64$ | – |
| 2 | Decoder | Unpooling | – | – | 0.3 (unpooling) |
| 2 | Decoder | SCE | $64 / 64$ | $64 * 2 / 64 * 2$ | – |
| 2 | Decoder | Conv+BN+ReLU | $128 / 128$ | $64 / 64$ | – |
| 1 | Decoder | Conv+BN+ReLU | $64 / 64$ | $32 / 32$ | – |
| 1 | Decoder | Unpooling | – | – | 0.3 (unpooling) |
| 1 | Decoder | SCE | $32 / 32$ | $32 * 2 / 32 * 2$ | – |
| 1 | Decoder | Conv+BN+ReLU | $64 / 64$ | $32 / 32$ | – |
| 1 | Final | Conv+BN | $32 / 32$ | $C / -$ | – |

Table F.5: Parameters of the architecture used for the segmentation experiments (cf Fig. F.6). $H_e$ denotes the number of attention heads in the encoder (3 in our experiments). $C$ is the number of classes in the dataset (4 for the categories `aliens` and `vases` of the COSEG dataset, 3 for the category `chairs` of the COSEG dataset, 8 for the Human Body dataset). $BN$ denotes batch normalization. $SC$ indicates a skip connection from the previous module. $SCE$ indicates a skip connection from the corresponding block in the encoder: the attention heads from the encoder are averaged and the feature is concatenated to the feature outputted by the decoder module. The attention heads are also averaged at the output of the single convolutional layer in Level 4.

**Superpixel-like segmentation**

We evaluate our method on the mesh segmentation task using also an encoder-only, alternative architecture, that we detail in the following. PD-MeshNet naturally forms clusters of faces through its task-driven pooling operation; therefore, by learning to form clusters of faces that all have the same ground-truth class label, the network could in principle predict a single class label for each cluster rather than a separate label for each face in the mesh. This idea relates to the concept of *superpixels* used in the context of image segmentation. Superpixels are sets of contiguous pixels that share common characteristics and that can be used to segment an image, by assigning them a class label. Usually, superpixels are formed through a clustering algorithm [28] based on pixel intensities and are later classified using learning-based techniques, but some works [29] have demonstrated the possibility of training a neural network to both form clusters of pixels and predict their class labels, in an end-to-end fashion. Similarly, we train PD-MeshNet to concurrently form clusters of faces – using its task-driven pooling operation – and label them, in an end-to-end fashion. We perform a small ablation study of this alternative method on the COSEG dataset. Fig. F.7 shows the *superpixel-like* architecture that we use in our additional evaluation. An encoder, made of 5 stacked residual convolutional blocks each with a single internal skip connection and each followed by a pooling layer, reduces the resolution of the input mesh by identifying face clusters (the mesh equivalent of superpixels in images). A final residual convolutional block predicts a class label for each cluster. The network is trained for 1000 epochs, using a fixed learning rate of $1e-3$ and optimizing a cross-entropy loss function computed on the labels of each of the faces of the input mesh. These are retrieved from the labels predicted on the face clusters by simply mapping each face of the input mesh to its corresponding cluster in the output mesh. Each convolutional block uses 3 attention heads; for the experiments on the `aliens` and `vases` categories, we contract $10\%$ of the primal edges in each pooling layer, while for the `chairs` category we set the fraction of primal edges to contract in each pooling layer to $5\%$. Dual-graph configuration Ⓐ is used. As shown in Tab. F.6,

Figure F.7: Superpixel-like architecture used for segmentation. An encoder, made of a series of stacked residual convolutional blocks each followed by a pooling layer, identifies clusters of faces in the input mesh (example from the COSEG dataset [27]). A final convolutional block assigns a class label to each cluster. The label of each face in the original mesh can be retrieved as the label of the corresponding cluster.

the accuracy on the test set is slightly inferior ($\sim 2.7\%$ to $\sim 3.3\%$) to the one obtained using the U-Net architecture (cf. Tab. 3 in the main paper). We believe that this difference in performance w.r.t. the U-Net architecture can be attributed to: (i) the lack of skip connections between the different blocks, (ii) more importantly, to the difficulty of the network in forming face clusters by contracting a predefined number of primal edges, without any auxiliary supervision on the formation of clusters. Possible future directions include adding an auxiliary loss to guide the formation of the clusters, incorporating skip connections between the blocks of the architecture, and investigating more flexible approaches w.r.t. the number of primal edges to collapse.

| Category | Face-label accuracy |
| --- | --- |
| aliens | 94.75% |
| chairs | 93.74% |
| vases | 92.79% |

Table F.6: Segmentation accuracy on the COSEG test dataset using the *superpixel-like* architecture.

## G  Ablation study on components

In the following, we evaluate the contribution of the components of our method by performing an ablation study. We use the Humany Body dataset with the same network parameters as the experiments from the main paper (cf. previous section), and train for 300 epochs, without data augmentation.

Table G.7 shows the result of the experiments. When pooling is removed, taking away the convolution on the dual graph (*Primal-only no Pool*) worsens performance by $\sim 31\%$ w.r.t. doing primal-dual convolution (*Ours no Pool*). Adding pooling significantly increases performance (by $6.74\%$, cf. *Ours* mean). We ablate the reduction function of the pooling layer and notice a small performance improvement when changing the aggregation from mean to sum (*Ours* add); for a more extensive evaluation of the influence of this factor, please cf. also Tab. H.8, which reports a comparison according also the edge-based labels.

| Primal-only no Pool | Ours no Pool | Ours mean | Ours add |
| --- | --- | --- | --- |
| 45.67% | 77.53% | 84.27% | **84.52%** |

Table G.7: Face-label accuracy on the Human Body dataset for the ablation study on the contributions of the network components.

Finally, we ablate the importance of the primal graph by removing pooling and predicting labels on dual nodes (mesh edges). Note that this accuracy is not comparable to the one in Tab. G.7. Using only the dual graph results in edge-label accuracy of $80.16\%$; on the other hand, adding the primal graph (while keeping the training/testing procedure still on edges) produces an accuracy of $80.39\%$.

## H  Metrics

For completeness, we report below the accuracy obtained by our approach according to two segmentation metrics that the method of Hanocka et al. [15] can be evaluated on, and that are different from the one based on face labels. The metrics below both require to predict class labels on the edges; therefore, it should be stressed that the results obtained on PD-MeshNet are not fully comparable with those obtained by [15], since our method predicts labels on faces, and the latter can at most be converted to *soft* labels on the edges, as shown in Fig. H.8. We denote the class label predicted on a generic mesh face $A$ as $l_A$, and the soft label of the edge between two generic adjacent mesh faces $A$ and $B$ as $l_{\{A,B\},\text{soft}}$. Since in an edge-manifold mesh each non-boundary edge is shared by exactly two faces, the soft label of the edge between two faces $A$ and $B$ can be expressed as a pair, with its element being the labels of the two faces $A$ and $B$, *i.e.*, $l_{\{A,B\},\text{soft}} = (l_A, l_B)$.

**Accuracy based on ground-truth edge *hard* labels.** This is the metric denoted as *Edge labels* in the main paper, and simply consists in evaluating the percentage of edges whose predicted label coincides with its single (*hard*) ground-truth label. Denoting the set of edges of a generic mesh $\mathcal{M}$ as $\mathcal{E}(\mathcal{M})$, the ground-truth label of a generic edge $\{A, B\} \in \mathcal{E}(\mathcal{M})$ as $l^*_{\{A,B\}}$, and its predicted class label as $l_{\{A,B\}}$, the accuracy can be found as:

$$\text{acc}_{\text{hard\_gt\_edge\_labels}} = \frac{\sum_{\{A,B\}\in\mathcal{E}(\mathcal{M})} [\![l_{\{A,B\}} = l^*_{\{A,B\}}]\!]}{|\mathcal{E}(\mathcal{M})|} \cdot 100, \qquad (16)$$

Figure H.8: Example conversion of face labels to edge soft labels. Assuming three possible class labels (corresponding to the red, green and blue colors on the left side), each edge is assigned a soft label by equally weighting the contributions of the two faces that share the edge (cf. right side).

where $[\![\cdot]\!]$ denotes the Iverson bracket. Since, as detailed above, the predictions of our method can only be converted to *soft* edge labels, for our approach we evaluate the accuracy (16) by separately assuming for each edge $\{A, B\} \in \mathcal{E}(\mathcal{M})$ the predicted *hard* label to coincide with the label predicted on the two faces $A$ and $B$, and equally weighing the two contributions, *i.e.*,:

$$\text{acc}_{\text{hard\_gt\_edge\_labels, ours}} = \frac{\sum_{\{A,B\} \in \mathcal{E}(\mathcal{M})} \left( 0.5 \cdot [\![l_A = l^*_{\{A,B\}}]\!] + 0.5 \cdot [\![l_B = l^*_{\{A,B\}}]\!] \right)}{|\mathcal{E}(\mathcal{M})|} \cdot 100. \quad (17)$$

**Accuracy based on ground-truth edge *soft* labels.** We further evaluate our method on the accuracy defined by [15]. This metric is based on ground-truth *soft* labels on the edges, and further weighs the contribution of each edge $\{A, B\} \in \mathcal{E}(\mathcal{M})$ by its length, which we denote as $\text{length}_{\{A,B\}}$. We refer the reader to the official code of [15] for the exact implementation details; for our purposes, we will just consider the accuracy as being, for each edge $\{A, B\}$, a generic function $f$ of the predicted *hard* label $l_{\{A,B\}}$, of the ground-truth *soft* label $l^*_{\{A,B\},\text{soft}}$, and of the edge length $\text{length}_{\{A,B\}}$. The metric can therefore be expressed as:

$$\text{acc}_{\text{soft\_gt\_edge\_labels}} = \frac{\sum_{\{A,B\} \in \mathcal{E}(\mathcal{M})} f \left( l_{\{A,B\}}, l^*_{\{A,B\},\text{soft}}, \text{length}_{\{A,B\}} \right)}{|\mathcal{E}(\mathcal{M})|} \cdot 100, \quad (18)$$

Similarly to (16), the problem with evaluating our method on the above metric is that the latter is a function, for each edge $\{A, B\}$ of the ground-truth *hard* label $l_{\{A,B\}}$. Also in this case, we therefore separately assume $l_{\{A,B\}}$ to coincide with the labels predicted on the two faces $A$ and $B$, and we equally weigh the two contributions, *i.e.*,:

$$\text{acc}_{\text{soft\_gt\_edge\_labels, ours}} = \frac{100}{|\mathcal{E}(\mathcal{M})|} \cdot \sum_{\{A,B\} \in \mathcal{E}(\mathcal{M})} \left[ 0.5 \cdot f \left( l_A, l^*_{\{A,B\},\text{soft}}, \text{length}_{\{A,B\}} \right) + \right.$$
$$\left. 0.5 \cdot f \left( l_B, l^*_{\{A,B\},\text{soft}}, \text{length}_{\{A,B\}} \right) \right]. \quad (19)$$

For the computation of the accuracy, we use the official code and ground-truth soft labels provided by [15]. We run our experiments using dual-graph configuration Ⓐ and the architecture detailed in Sec. F.2. Similarly to the main paper, we run each of the experiments of [15] 4 times – using the official code and parameters provided – and we report the best results obtained.

| Dataset | Method | Metric | | |
|---|---|---|---|---|
| | | Edge labels (hard ground-truth) | Edge labels (soft ground-truth) | Face labels |
| COSEG aliens | MeshCNN [15] | 95.35% | 97.14% | 96.26% |
| | Ours mean | 96.51% | 98.53% | 97.63% |
| | Ours add | **97.06%** | **99.03%** | **98.18%** |
| COSEG chairs | MeshCNN [15] | 92.65% | 94.66% | 92.99% |
| | Ours mean | 96.26% | 97.93% | 97.08% |
| | Ours add | **96.44%** | **98.21%** | **97.23%** |
| COSEG vases | MeshCNN [15] | 91.96% | 96.91% | 92.38% |
| | Ours mean | **94.70%** | **97.96%** | **95.47%** |
| | Ours add | 94.57% | 97.83% | 95.36% |
| Human Body | MeshCNN [15] | 84.05% | **92.05%** | 85.39% |
| | Ours mean | 84.13% | 90.44% | 84.78% |
| | Ours add | **85.09%** | 91.11% | **85.61%** |

Table H.8: Test accuracy on the mesh segmentation task according to the metrics defined in the paper. "Ours mean" and "Ours add" denote our method respectively with averaging and summation pooling aggregation, cf. Sec. 3.3 in the main paper.

## I  Runtime

We perform our experiments using a single NVIDIA Quadro RTX 8000 GPU. Using the architectures detailed in Sec. F, one training epoch on the Split 16 of the SHREC dataset takes approximately 70s, while one training epoch on the aliens category of the COSEG dataset lasts for around 64s. The average time for a forward pass on a single mesh (*i.e.*, batch size 1) is $\sim 175$ms for the experiment on the SHREC dataset, $\sim 390$ms for the one on the COSEG aliens (1500 faces per mesh) dataset, and $\sim 279$ms for the experiments on the COSEG chairs/vases (1000 faces per mesh) dataset.