[Reviews · NeurIPS 2020]

Review 1

Summary and Contributions: This paper addresses a mesh-based deep CNN architecture based on primal-dual meshes, and task-specific pooling. The paper shows improved performance on shape classification and shape segmentation tasks.

Strengths: The overall approach is well designed. The method shows improved performance.

Weaknesses: It is unclear what the contributions of the paper are, apart from combining various techniques: primal-dual framework is not new, dynamic aggregation for meshes is not new (e.g. MeshCNN), and attention maybe new for mesh convolution, but is widely used in most topics with neural networks. The paper should avoid overclaims. The paper says "These methods, however, either consider the input mesh as a graph, and do not exploit specific geometric properties of meshes for feature aggregation and downsampling, or are specialized for meshes, and rely on a rigid definition of convolution that does not properly capture the local topology of the mesh." However, the proposed method is also specialized for meshes. Yes, it considers two graphs (primal, dual meshes) but that's not much different from one graph. The paper later says "the unique feature of our approach is the geometric interpretation of this operation (pooling)". Yes, your method may have an easier geometric interpretation, but it is hard to say that existing methods do NOT have geometric interpretation. Another weakness is the proposed method is much more complicated than alternative methods, for example MeshCNN. However, the improvement is rather limited, and it is not clear the benefits of individual components. Ablation study is not provided in the paper, and the supplementary material only contains a rather limited ablation study that is not sufficient.

Correctness: The method seems correct, but claims should be carefully revised and contributions clearly stated.

Clarity: The writing is acceptable.

Relation to Prior Work: Partially, but the technical contributions should be carefully reworded.

Reproducibility: Yes

Additional Feedback:


Review 2

Summary and Contributions: This paper introduces a novel neural network architecture processing 3D meshes. The main idea is to apply a recent primal-dual graph convolution technique introduced in [20] to meshes, but the authors also integrate mesh-specific components in the pipeline such as using geometric features as input and employing edge contraction as a pooling operation, as similarly done in [12]. The experiments demonstrate the STOA performance of the proposed method in various classification and segmentation tasks. Also, the supplementary document provides vast details about technical differences with previous work, convolution and pooling operations, network training, and possible other configurations of the dual graphs (along with experimental analyses).

Strengths: - SOTA performance in 3D mesh classification/segmentation. - Extensive and detailed expositions --- I especially appreciate the thorough analyses and discussions in the supplementary document.

Weaknesses: - Incremental approach --- the main technical contribution is in combining basic technical components coming from two papers: Monti et al. [20] and Hanocka et al. [12].

Correctness: Mostly yes, but please see the comments in the additional feedback.

Clarity: Yes

Relation to Prior Work: Yes

Reproducibility: Yes

Additional Feedback: While this paper looks a bit like an incremental extension of the primal-dual graph convolution [20] to a mesh-specific case (using the ideas of MeshCNN [12]), I would like to recommend acceptance due to its comprehensive and detailed expositions and analyses. The paper is well-written to see its differences/advantages compared to the other related work and also understand all the technical details. I also like the authors’ considerations for variants of the proposed method (the different ways of building primal-dual graphs in the supplementary document). Despite the details expositions, I have some questions/comments: - I'm not sure about the duality of graphs P(M) and D(M). To my knowledge, duality means that the function converting one to the other can also return back to the other way around. The graphs M and P(M) are dual (as mentioned in the paper) in this sense since the way going from M to P(M) (taking faces as vertices) also converts P(M) to M. This rule does not hold for P(M) and D(M). I guess the notions of primal and dual graphs for P(M) and D(M) may not be technically correct? - While the authors take the simplex mesh P(M) as the primal graph in all of their primal-dual configurations, I think another primal-dual configuration taking the input mesh M itself as the primal graph can also be considered. I guess the main reason why the authors take P(M) as the primal graph would be to predict per-face information in segmentation and also directly use the geometric features introduced in MeshCNN [12]. But if one needs to predict per-vertex information, the other configuration would be useful. - Supplementary line 36: "triangle mesh M the medial graph" -> What does this mean? - Supplementary line 100: "in alternative to (4)" -> in alternative to (2)? ************** After Rebuttal ************** I read the rebuttal and decided to hold my decision to accept.


Review 3

Summary and Contributions: A mesh cnn method that leverages geometric reasoning and the dual-primal graph convolution method from [20]. The Primal and Dual are built from the input mesh; generalizing the work from Hanocka et al. Good performance is demonstrated on classification and segmentation tasks.

Strengths: + the paper is very well written and easy to follow. Very helpful details and illustrations are provided, aiding in fully appreciating the idea. + The primal-dual graph method was not shown before for mesh processing. Integrating it with geometric features is quite elegant + thorough experimentation is providing evidence for improved performance on a variety of tasks.

Weaknesses: - It is not clear to me wether the pooling is done sequencially or in parallel. - The unpooling issue of missing nodes was not clear. I'd ask the authors to expand more about that - When pooling, it seems that feature averaging was done. Were any other pooling methods tried? - What are other limitations of the method? in the graph case the network was pretty shallow, is this the case here?

Correctness: Yes

Clarity: Yes, very well

Relation to Prior Work: * Along with scene mesh data i would include datasets like FAUST and DynamicFAUST which encouraged lots of recent DL for meshes works. * SpiralNet++: A Fast and Highly Efficient Mesh Convolution Operator * A simple approach to intrinsic correspondence learning on unstructured 3d meshes * Dynamic graph cnn (DGCNN)

Reproducibility: No

Additional Feedback: After reading the rebuttal and other reviews I think the main debate is whether putting together the two ideas of primal-dual for general graphs; with mesh specific cnn is novel enough. I'm pretty convinced that it is, especially since it's done well by the authors. I would be happy to see it in the conference.


Review 4

Summary and Contributions: This paper introduces a novel scheme to learn from meshes by reasoning on both a primal graph, defined on the faces of the meshes, as well as its dual. The authors claim that such a formulation exploits both the geometric properties and the local topology of a mesh. Empirical evidences seem to support this claim to a limited extent.

Strengths: -Decent theoretical motivation of the primal-dual formulation. -Solid analysis of limitations of prior works, and why the proposed method supposedly resolves those.

Weaknesses: -Lack of justification on why the proposed method actually helps. The reasons given for the improved results in Section 4 is way too speculative, I expect to see more detailed analysis supporting claims such as "capturing high-level structural information" (L285), "aggregating semantic information" (L318), etc. -Even assuming that those claims are true, the improvements do not seem to be significant enough to me, especially considering that some comparison numbers are taken directly from [12] (Table 1 & 2) but some seems to be re-ran and lower than those shown in [12] (Table 3 & 4)

Correctness: The method is based on known results regarding mesh operations and regarding primal-dual graph convolution. I think the authors combine those correctly.

Clarity: The paper is in general well written and easy to understand.

Relation to Prior Work: As far as I am aware, the authors have discussed the most relevant works in the field and explained the relations of them to this work. However, I do feel that the differences between this work and prior works are not as large as what the authors seem to claim here.

Reproducibility: Yes

Additional Feedback: Overall, I tend to reject this paper (5 leaning towards 4) for two reasons: 1. that there is very limited novelty, using the model proposed in [20] and the features proposed in [12]. 2. that the claimed advantages of the work is unsubstantiated: the authors only show that their method outperforms [12] in a set of (questionable) comparisons, but never explained why their new formulation is responsible for such improvements. I will be willing to adjust my rating if the authors could convince me that their method is actually critical to the performance increases (this invalidates 2. while making 1. a much lesser concern). Additional comments: - I am not sure if the proposed architecture leverages any properties of a (manifold or not) mesh. It seems that this primal-dual formulation is applicable to any graphs, despite that the fact that some operations might be equivalent to certain mesh operations. How is the method different from graph convolutional networks on arbitrary graphs? If not, why is using a general architecture on a specific domain a valid contribution (in other words, why is this "knowledge transfer" a very valuable one that will change the domain of mesh analysis) -(L6) could the authors elaborate why methods such as MeshCNN "does not properly capture the local topology of the mesh" but PD-MeshNet, defined on faces, does? -As mentioned in earlier sections, the authors attributed the better performances to the model's ability to "capturing high-level structural information" (L285), "aggregating semantic information" (L318), etc. However, unless I'm missing something here, the authors didn't really provide insights on why this is the case, nor did the authors provide any ablations (I don't think the one on dual graph types count) here. I expect the authors to provide justifications for those, since those are very important claims. -I have problems with the evaluations numbers in Table 1-4. It seems that the numbers in Table 1 and 2 are taken directly from the MeshCNN paper whereas the numbers in Table 3 and 4 are not (and are lower than those in the MeshCNN paper). Could the authors explain the reasons behind this decision? -(L325) "We attribute this result to the limited size of the test dataset..." Why is this only a problem for PD-MeshNet but not MeshCNN? -Figure 6: I get that the segmentation results look good, but are those better than prior works? =====Post-rebuttal comment===== I tried to reproduce the MeshCNN results and would acknowledge that the author's comparisons are indeed fair. The discussions between reviewers boil down to the question whether applying an existing approach to a new domain is a novel enough contribution. This question is quite objective, and I can understand why other reviewers lean on the more positive side. However, I still hold my opinion that this work would be much better if the authors can *show with evidence* that their method indeed displays those advantages they claim. (see my reviews earlier). I will keep my score - I'm definitely more positive post rebuttal though.

[Author Response · NeurIPS 2020]

1. We thank the reviewers for their valuable feedback. We are pleased that they found the approach to be well-designed
2. [**R1**], the expositions, illustrations, and experimentation to be comprehensive and detailed [**R2**, **R3**], the method "quite
3. elegant" [**R3**], and the analysis of limitations of prior works solid [**R4**].
4. **Contributions.** We apologize for the lack of clarity that caused some sentences to be perceived as overclaiming;
5. we will carefully rephrase them. **R1**, **R2**, **R4**: *"incrementality of approach"/"value of knowledge transfer"*: while
6. it is true that we use the convolution from [20], we would like to stress that this was only applied on generic
7. graph benchmarks, and that the idea of reconducting primal-dual graphs to meshes is novel. This extension, to-
8. gether with the idea of "bridging the gap" between graph-based ([20]) and ad-hoc methods for meshes, enables
9. to use tools from the graph-NN literature in the context of meshes and to combine them with tools from the
10. mesh-processing literature. Specifically, it allows implementing dynamic aggregation on meshes (via attention),
11. and to assign features not only to edges, but also to faces. Finally, we are the first, in the context of learning-
12. based mesh processing, to perform an attention-based pooling operation, which can be geometrically interpreted as
13. face clustering. **R2**: *"geometric interpretation"*: the "uniqueness" is in its *type*, not in the fact that we have one;
14. **R4**: *"network identifies high-level structure with semantic meaning"*:

15. we apologize for the lack of clarity: our intention was not to claim that
16. all sub-parts identified by the network have a semantic meaning, but
17. rather that the approach can identify «semantic components in a mesh
18. *more naturally* [...]» than other methods. (L318-320). We believe this
19. work paves the way for a number of research avenues (L334-337): our pooling operation goes in the direction of
20. abstracting similar larger structures across similar samples. In the Figure above (same color ↔ same face cluster):
21. on the left, the legs of the chairs are identified as clusters; on the right, larger planar structures are found. We stress
22. that the network does not always find such structures, but also highlight that no supervision on face clusters was
23. provided during training. To reproduce figures: see PD-MeshNet/docs/results.md, command python test.py
24. --f $UNZIP_FOLDER/coseg/chairs/ --save_clusters.
25. **Ablation studies.** As suggested [**R1**, **R4**], we provide ablation
26. studies to evaluate the contribution of the components. We
27. use the Human Body dataset with same network parameters as
28. the main paper, and train for 300 epochs. We do not perform
29. data augmentation to further isolate the importance of each

| P no Pool | Ours no Pool | Ours (mean) | Ours (sum) |
|-----------|--------------|-------------|------------|
| 45.67% | 77.53% | 84.27% | **84.52%** |

Table 1: Ablation Study (Face-label acc.)

30. component. Table 1 shows the result of the experiments. When pooling is removed, taking away the convolution on the
31. dual graph (*P no Pool*) worsens performance by $\sim 31\%$ w.r.t. doing primal-dual convolution (*Ours no Pool*). Adding
32. pooling significantly increases performance (by $6.74\%$, cf. *Ours (mean)*). Following the suggestion of **R3**, we ablated
33. the reduction function of the pooling layer and noticed a very small performance improvement when changing the
34. aggregation from mean to sum (*Ours (sum)*). Finally, we ablate the importance of the dual-graph by removing pooling
35. and predicting labels on dual nodes (mesh edges). Note that this accuracy is not comparable to the one in Tab. 1. Using
36. only the dual graph results in edge-label accuracy of $80.16\%$; on the other hand, adding the primal graph (while keeping
37. the training/testing procedure still on edges) produces an accuracy of $80.39\%$.
38. **Improvement over SOTA.** We acknowledge that our method is slightly "more complicated" [**R1**] than others, but we
39. would like to stress that we improve on the SOTA by up to $8.1\%$ on SHREC, $2.23\%$ on Cube Engraving and up to
40. $5\%$ on COSEG (cf. Table $1 - 3$ in main paper). **R4**: *"different metrics"/"re-run results"*: MeshCNN uses an ad-hoc
41. accuracy for segmentation; please cf. Sec. G of supplementary for more details. To ensure fairness, Table G.7 in
42. the supplementary reports results for 3 different accuracies, including the one by MeshCNN. Furthermore, while for
43. classification we were able to reproduce the results of MeshCNN, this was partially not the case for COSEG (the
44. experiment on the chairs subset gave results significantly lower than those reported in their paper). We thus decided
45. to report the values that we were able to reproduce with the official MeshCNN code.
46. **Technical details and clarifications.** **R2**: *"duality"*: $\mathcal{G}(\mathcal{M})$ and $\mathcal{P}(\mathcal{M})$ are dual in the classical graph-theory sense
47. (e.g., [33, 34]), while $\mathcal{P}(\mathcal{M})$ and $\mathcal{D}(\mathcal{M})$ are dual in the sense introduced by [20]; *"$\mathcal{G}(\mathcal{M})$ as primal graph"*: possible
48. extension, but using $\mathcal{P}(\mathcal{M})$ as primal graph also allows defining our pooling operation: edge contraction in $\mathcal{G}(\mathcal{M})$ is
49. instead a classical "edge collapse", as in MeshCNN, thus more limited (cf. Sec. C.1 in supplementary); **R3**: *"pooling"*:
50. it is done in parallel through tensor operations; **R4**: *"size of test dataset"*: not a problem only for our method; we wanted
51. to stress that the limited size of the test dataset and the «the large intra-class variability» (L326) could amplify the effect
52. of wrong predictions; **R4**: *"capturing the local topology of the mesh"*: the reason is in the convolution: MeshCNN
53. aggregates features weighted by kernels *shared by all the mesh edges*, thus not tuned on the local region on which they
54. are applied. The convolution of our approach ([20]) implements instead an attention mechanism, by which nodes in
55. different regions of the mesh can aggregate information differently, depending on the local properties of the mesh.
56. **Limitations.** [**R3**] We indeed empirically noticed a performance drop when the network is too deep (due to the larger
57. number of parameters associated with the two graphs and the more complex optimization landscape). Moreover, pooling
58. layers can be sensitive to their threshold parameters. Indeed, too-aggressive pooling causes a degradation of results.
59. **Miscellaneous.** We will add a discussion on the additional related work [**R3**] and correct typos [**R2**].

[Meta-Review · NeurIPS 2020]

Two referees are very positive about this paper and recommend acceptance, whereas two referees lean towards rejection. All referees agree that primal-dual graph networks have not been previously investigated for mesh processing, but disagree on whether the contribution should be considered significant and/or incremental. The rebuttal attempts to address this concern by highlighting that primal-dual approaches have only been applied on generic graph benchmarks and further emphasizing that the application of such approaches to meshes is not straightforward and only constitutes one part of the proposed approach. R1 and R4 raised concerns about the experimental validation. The rebuttal only partially addressed their concerns, but after discussion, R4 is convinced that the experimental validation is compelling enough. However, R4 still finds the contribution largely incremental. I agree with the overall assessment that this is a well executed contribution with compelling and comprehensive experimental validation, and I support R2 and R3's opinion that borrowing an existing approach to strengthen another approach may still be a valid contribution. Therefore, I recommend acceptance.